# MCF7 Spheroid Development: New Insight about Spatio/Temporal Arrangements of TNTs, Amyloid Fibrils, Cell Connections, and Cellular Bridges

**DOI:** 10.3390/ijms21155400

**Published:** 2020-07-29

**Authors:** Laura Pulze, Terenzio Congiu, Tiziana A. L. Brevini, Annalisa Grimaldi, Gianluca Tettamanti, Paola D’Antona, Nicolò Baranzini, Francesco Acquati, Federico Ferraro, Magda de Eguileor

**Affiliations:** 1Department of Biotechnology and Life Sciences, University of Insubria, via J.H. Dunant 3, 21100 Varese, Italy; l.pulze@uninsubria.it (L.P.); Annalisa.grimaldi@uninsubria.it (A.G.); gianluca.tettamanti@uninsubria.it (G.T.); p.dantona@uninsubria.it (P.D.); n.baranzini@uninsubria.it (N.B.); francesco.acquati@uninsubria.it (F.A.); federico_ferraro@tiscali.it (F.F.); 2Department of Surgical Science, University of Cagliari, via Università 40, 09124 Cagliari, Italy; terenzio.congiu@unica.it; 3Laboratory of Biomedical Embryology, Centre for Stem Cell research, University of Milan, via Celoria 10, 20133 Milan, Italy; tiziana.brevini@unimi.it

**Keywords:** MCF7 3D spheroid, TNTs, amyloid fibrils, gap junctions, cytoplasmic bridges, ROS scavenger, NEP treatment

## Abstract

Human breast adenocarcinoma cells (MCF7) grow in three-dimensional culture as spheroids that represent the structural complexity of avascular tumors. Therefore, spheroids offer a powerful tool for studying cancer development, aggressiveness, and drug resistance. Notwithstanding the large amount of data regarding the formation of MCF7 spheroids, a detailed description of the morpho-functional changes during their aggregation and maturation is still lacking. In this study, in addition to the already established role of gap junctions, we show evidence of tunneling nanotube (TNT) formation, amyloid fibril production, and opening of large stable cellular bridges, thus reporting the sequential events leading to MCF7 spheroid formation. The variation in cell phenotypes, sustained by dynamic expression of multiple proteins, leads to complex networking among cells similar to the sequence of morphogenetic steps occurring in embryogenesis/organogenesis. On the basis of the observation that early events in spheroid formation are strictly linked to the redox homeostasis, which in turn regulate amyloidogenesis, we show that the administration of *N*-acetyl-l-cysteine (NAC), a reactive oxygen species (ROS) scavenger that reduces the capability of cells to produce amyloid fibrils, significantly affects their ability to aggregate. Moreover, cells aggregation events, which exploit the intrinsic adhesiveness of amyloid fibrils, significantly decrease following the administration during the early aggregation phase of neutral endopeptidase (NEP), an amyloid degrading enzyme.

## 1. Introduction

The full understanding of the molecular mechanisms underlining cell–cell interactions, as well as the specific molecular events that participate in their dynamic changes in breast cancer formation and growth, is of paramount importance. To improve the accuracy of solid tumor-based research models in order to decipher the underlining pathways, and to better describe the morpho-functional changes that determine the histological complexity of this solid tumor, three-dimensional cell culture system have been extensively used in the last decade [1,2,3,4,5,6,7].

The most recent studies point at three-dimensional (3D) spheroids as a good model for basic biological research on multicellular tumors [8,9,10,11,12] as well as for selection of new anti-cancer drug [13,14,15]. Cells spheroids gained a growing attention thanks to their ability to **mimic** in vitro avascular tumor, characterized by heterogeneous population of cells and, as such, represent a powerful tool to study their development, morphological changes, and metabolism [16]. At the cellular level, a large amount of data regards intercellular communications as critical points for tumor formation, development, and treatment resistance [17]. Moreover, current research on different cancer cells (from lung, pancreas, prostate, bladder, nervous system) shows that tunneling nanotubes (TNTs), exosomes, and gap junctions are involved in this intercellular network [17,18,19,20,21,22,23,24].

TNTs are thin, long nano-canals that stretch towards neighboring cells [19]. These structures are easily detected in cultured cells, in developing embryos, and in patients’ solid tumors, and are functionally relevant entities that facilitate direct communication between distant cells, by allowing cell-to-cell transfer of electrical and mechanical signals or cellular contents [20,21,22,24], thereby regulating the cell’s differentiation state [18,20].

Exosomes and macrovesicles are small membrane vesicles delivering protein, lipids, and nucleic acids, and are significantly involved in the intercellular cross talk [18,23]. In the context of cancer biology, these carriers are apparently able to modulate a wide range of signaling pathways [17,21]. In the context of cancer cell-to-cell communication, ample evidence shows the critical role of gap junctions and their protein subunits (connexins) in directly linking the cytoplasm of adjacent cells, coupling them electrically [18].

Notwithstanding the large amount of data regarding the formation and development of spheroids, currently, to our best knowledge, only scattered information is available on the morpho-functional characterization of the different phases leading to human breast adenocarcinoma cells (MCF7)-based mature spheroid.

Here, we report a detailed analysis of the pattern of human MCF7 spheroid formation and we describe different phenomena responsible for the reciprocal adhesion in the early and last phases. Moreover, we describe the strategies adopted to assemble a compact cellular globe based on physiological cellular processes.

When cultured in ultra-low attachment plates [25], MCF7 cells form small aggregates within 24 h and, starting from approximately 3 days after seeding, evolve into spheroids. In the earliest aggregation steps, the cells interact via TNTs and, strikingly, facing cells physiologically produce amyloid material [26,27,28,29]. Amyloid fibrils, owing to their intrinsic mechanical and biological properties, allow the maintenance of an indirect contact providing the driving force for early aggregation [27]. These two early events are strictly linked to the increase of reactive oxygen species (ROS) in the cytoplasm. It is well established that ROS behave as early inducers and regulators of a plethora of events such as cell growth and differentiation, cell death, initiation, and progression of cancer [30]. Moreover, as ROS-exposed proteins adopt a β-sheet conformation, ROS are closely related to synthesis of amyloid [31,32], and the formation of membrane protrusions (TNTs) [22].

Moreover, gap junction formation synergistically concurs to spheroid development.

In later phases of spheroid maturation, an additional cell-to-cell-communication mechanism is called into play, based on the formation of large stable cytoplasmic bridges that allow a movement and exchange of molecules and organelles [33,34]. Moreover, these close connections induce the spheroid to functionally behave as a syncytium, thus explaining the observed increased mechanical resistance of the whole spheroid reported by Ivascu and Kubbies [35].

This study, in particular the detailed description of the processes driving spheroid aggregation, the observed role of redox homeostasis, and the functional link between amyloid fibril disruption leading to the spheroid growth pattern, sheds a further light on breast-cancer biology and might pave the way for the development of new treatments.

## 2. Results

### 2.1. Spheroid Formation: Morphology and Behaviour of MCF7 Cells

As expected, when plated onto ultra-low attachment plates, MCF7 cells rapidly formed 3D spheroids [8,9,10]. Time frames were selected to highlight the most significant events leading to the maximum development of the spheroid.

Initially, cells achieved several reciprocal close contacts in about two weeks, starting from tiny aggregates that were formed by few cells and later forming large globes showing a different morphology.

Twenty-four hour spheroid culture: during the first few hours of cell culture, cells were roundish or oval in shape (Figure 1A–D). Scanning electron microscopy (SEM) images showed cells exhibiting a few thin, long cytoplasmic projections resembling TNTs (Figure 1B). During this period, cells formed aggregates that were different in size (Figure 1C,D) with an observed average of 46.92 ± 1.10 μm in diameter (Figure 1A).

Most cells were maintained in close reciprocal proximity through intertwined microvilli (Figure 1E) and by a structured dense matrix that was released by the cells and distributed over their surface (Figure 1C–E). Transmission electron microscopy (TEM) analysis confirmed the presence of TNT-like structures, and showed that the cytoplasm of MCF7 cells was characterized by large nuclei, Golgi apparatus, mitochondria, vacuoles, and enlarged rough endoplasmic reticulum (RER) cisternae filled with fibrillar material, whose presence is generally linked to stress conditions [28,29,36] (Figure 1F–H). Both intracellular multi-vesicular bodies and released exosomes were evident in the space separating independent cells (Figure 1I).

Three day spheroid culture: MCF7 cells assembled in compact small spheroids of 84.67 ± 4.88 μm in diameter (Figure 1J). SEM and TEM analyses highlighted the different phenotypes of aggregated cells (Figure 1J–O). The majority of the cells were polygonal in shape and in close reciprocal contact, by adhering together without extracellular matrix interconnections (Figure 1J,L,M). Very few specialized connections were visible by TEM analysis (Figure 1L,M). The adhesion and consequent compaction pattern were also correlated to the clear polarization of the outer cells that were characterized by microvilled outward-face and contiguous baso-lateral faces (Figure 1M). Only these outermost cells, representing those interfaced with the external environment, showed dilated RER cisternae filled with the material destined to be exocytosed (Figure 1N,O).

Five–seven day spheroid culture: cell spheroids grew over time by increasing in size, with a mean diameter of 163.38 ± 10.99 μm (Figure 2A). As observed by both SEM and TEM analyses, the spheroids were made of an increased number of cells, different in shape, size, and type of connections (Figure 2A–G). The nucleus–cytoplasm ratio was high (a hallmark of stem cells) and many cells showed the presence of euchromatic nucleus with large nucleoli, reflecting intense metabolic activity (Figure 2D). Cells were in close contact and interconnected (Figure 2B–F). Moreover, an unambiguous, ultrastructural profile represented by gap junctions was observed (Figure 2C,F). The cell phenotype changed also owing to the presence of filament bundles within the cytoplasm and under the plasma membrane (Figure 2G).

Eleven–sixteen day spheroid culture: these large multicellular spheroids reached a mean of 270.54 ± 25.22 μm in diameter (Figure 2H). They were formed by cells showing various stages of differentiation and characterized by variable sizes (Figure 2H–L). Moreover, electron dense clusters of assembled fibrous proteins (Figure 2K) and stable intercellular bridges were clearly detected (Figure 2L,M).

These large cytoplasmic passageways, displaying a diameter of about 2 μm, joined neighboring cells and sometimes were so large that it was difficult to identify the primitive cell borders (Figure 2L,M).

The increasing average size of the spheroids was relatively uniform and synchronous in their formation. Starting from 24 h up to 16 days of culture, the mean diameter doubled stepwise from one time point to the next, as validated by statistical analysis (Appendix A).

### 2.2. Closely Related Events During Spheroid Maturation

#### 2.2.1. Early TNT Formation

As shown in Figure 1A,B,F,G and Figure 3A–C, the dynamic and transient structures represented by TNTs [19,24] allowed the cytoplasmic connections and reflected the spatial position of cells that were far apart (Figure 3A–C).

The presence of 30 μm long TNTs with a diameter up to 1 μm, as detected ultrastructurally, was further supported by the expression in cells of testis expressed gene 14 (TEX14), a protein involved in cellular (dynamic and stable) bridge formation (Figure 3B) and by the presence of actin filaments typically disposed along these cytoplasmic extensions (Figure 3C).

The different expression levels of TEX14 were validated by Western blot analysis (Figure 4).

#### 2.2.2. Synthesis of Amyloid Fibrils

Given the ultrastructural similarities between the fibrillar material present in the ER of different cell types [26,27,28,29] and that observed in aggregating MCF7, we investigated whether amyloidogenesis, sustained by changes of cytoplasmic redox potential, could be involved in MCF7 spheroid formation as well.

Amyloid fibrils, initially located in the dilated cisternae of RER and in the space among cells (Figure 1E,H), were evaluated for both their cross-β-sheet core and for their constituent protein content (Figure 3D–H). The amyloid material showed a typical thioflavine S (ThS) positive staining (Figure 3D,G,I). The characteristic bright yellow-green fluorescence was detected both in the cytoplasm and in the intercellular space, more readily evident in cells located at the external side of the small spheroid or at the internal side when cells were distanced (i.e., amyloid material is exocytosed principally during the early phase of spheroid formation).

The presence of amyloid was further confirmed by the immunolocalization assays with a specific antibody raised against the melanocyte protein (Pmel17), a mammalian protein involved in amyloidogenesis [26,27,28,29] (Figure 3E,F,H,J,K). The signal was detectable in the same areas that were also ThS positive. The different expression levels of Pmel17 were also validated by Western blot analysis (Figure 4).

#### 2.2.3. Intracellular ROS Evaluation

The overall intracellular level of ROS, detected using the fluorigenic probe 2′,7′-dichlorodihydrofluorescein diacetate (H2DCFDA), persisted during the various developmental phases of spheroid growth (Figure 3L–P).

#### 2.2.4. Expression of Stemness Markers

Cells from early-phase spheroids (from 24 h up to 7 days) expressed stemness markers (Figure 3Q–Z) such as the ganglioside stage-specific embryonic antigen-4 (SSEA-4) and the SRY (sex determining region Y)-box 2 (Sox2) proteins [6,37,38] (Figure 3Q–S,V–X), whereas lower expression levels were detected in the last phase of spheroid maturation, when cells displayed a more mature phenotype (Figure 3T,U,Y,Z). Validation of Sox2 levels by Western blot analysis showed that its expression peaked in correspondence with dimensional increase of spheroids and then it decreased in the last phase of development (Figure 4).

#### 2.2.5. ACTH/α-MSH Axis Activation

ACTH/α-MSH expression (Figure 5A–H), was easily detected by immunocytochemical assays performed at both 24 h (Figure 5A,E) and 3–5 days (Figure 5B,F) stages.

By contrast, in mature spheroid, expression of ACTH/α-MSH was mainly confined to the external cells (Figure 3C,D,G,H).

#### 2.2.6. Interleukin-18 (IL18) Production

MCF7 cells aggregation and compaction leading to the assembly of 3D spheroids were accompanied by the production of pro-inflammatory cytokines. Among the plethora of these cytokines, IL18 was examined owing to its link with ROS generation and amyloid deposition [28,29,39]. The regional distribution of this marker was assessed by immunocytochemistry and detected between the faces of those cells that were kept close by the adhesive action of the amyloid fibrils (i.e., in early aggregates) (Figure 5I,J). By contrast, IL18 expression was reduced and localized in the superficial cells in mature spheroid (Figure 5K,L). These data were also confirmed by Western blot (Figure 6) and by a quantitative ELISA assay to detect the actively released protein (Figure 6).

#### 2.2.7. Cell–Cell Connections

Morphological analysis demonstrated the presence of specific sites of intercellular connections. Starting from early aggregation (temporal frame: 24 h/3 days), the cells showed discontinuous areas where plasma membranes of adjacent cells were in adhesive contact (Figure 1H,L–N).

In a stepwise sequence over the 3 day period, cells were pressed together and gap junctions could be identified (Figure 2F and Figure 5M). Starting from the aggregation phase until the formation of compact spheroids, E-cadherin and Connexin 43 were expressed, as demonstrated by immunolocalization assays (Figure 5N,O). In particular, E-cadherin, an integral component of gap junctions, showed a typical punctate staining (Figure 5O). The increased expression linked to the plasticity and asynchronous differentiation processes of spheroids-forming cells was also validated by Connexin 43 expression analysis by Western blot (Figure 6). In addition, the different levels of cell differentiation in well-formed spheroids were evaluated by double labelling assays showing that, in cells displaying enhanced expression of Sox2 stemness markers, the distinguishing features of differentiation (E-cadherin) were attenuated or absent, and vice versa (Figure 5P).

Among cell–cell connections, stable large intercellular bridges (Figure 5Q,R) represent essential components. In this context, TEX14 expression was confirmed by immunofluorescence staining (Figure 5S) and by Western blot (Figure 4).

#### 2.2.8. Synthesis of Actin and Cytokeratin Filaments

Actin filaments (Figure 7A–D), being crucial cytoplasmic structural components, assure stability and resistance to mechanical forces [40]. Actin bundles were thus assessed ultrastructurally and shown to be increased from early to mature spheroids, as validated in fluorescence microscopy (Figure 7E–H).

Cytokeratin filaments were morphologically evidenced in the well-differentiated cells from 5–7 to 11–16 days spheroids (Figure 7I,J). Cytokeratins, expressed at various stages of development and differentiation, were identified using a pancytokeratin antibody (Figure 7K–N) to evaluate the differentiation state [41].

### 2.3. Functional Studies

Functional activities of MCF7 cells were found to drastically change, especially in the early and last phases of spheroid formation.

#### 2.3.1. Detoxifying Agent: NAC Treatment

During the early phase of cell aggregation, amyloid fibril synthesis was observed. On the basis of the documented relationship between amyloidogenesis and changes in the cell’s redox status (cytoplasmic pH modification) [32,42], we administered a molecule acting as an ROS scavenger to indirectly influence amyloid fibril synthesis (Figure 8A–I).

In cells treated with NAC, the reduced production of adhesive amyloid material led to a decrease in the rate of MCF7 cellular aggregation at 24 h (mean diameter of 28.28 ± 0.43 μm) (Figure 8A,B), as compared with controls (mean diameter of 46.82 ± 1.46 μm) (Figure 8F,G). Cells could be characterized by cytoplasmic protrusions (Figure 8B).

NAC-treated MCF7 cells at 3 days (mean diameter of 60.29 ± 2.63 μm) (Figure 8C–E) were smaller than controls (mean diameter of 87.65 ± 3.75 μm) (Figure 8H,I). The reduced ability to maintain the aggregation state was attributable to the detachment of spheroid external cells (Figure 8E). Data about NAC treatment were validated by quantitative analysis (Appendix A).

#### 2.3.2. Amyloid-Degrading Enzyme: NEP Administration

To confirm the physiological role of the amyloidogenesis in the aggregation process (early stages), we disrupted the amyloid fibril-assemblage by NEP treatment. This enzyme is known to massively hydrolyze amyloid fibrils, thereby preventing their accumulation [43,44,45,46]. NEP treatment drastically reduced the ability of the cells to aggregate by exploiting the intrinsic adhesiveness of amyloid fibrils. Indeed, starting from 24 h after seeding, most cells lost the ability to remain in close contact (Figure 8J,K). Compared with controls (mean diameter of 46.82 ± 1.46 μm) (Figure 8F,G), NEP-treated aggregates were drastically reduced in size (mean diameter of 31.68 ± 0.40 μm) (Figure 8J,K). Following 3 days of NEP treatment (Figure 8L,M), aggregates were significantly smaller (mean diameter of 59.09 ± 1.99 μm) than controls (mean diameter of 87.65 ± 3.75 μm) (Figure 8H,I) and showed irregular profiles (Figure 8L,M). Data about NEP treatment were validated by quantitative analysis (Appendix A).

#### 2.3.3. Fluorescent Dextran Microinjection

During the last step of spheroid development, functional trafficking activity through large stable intercellular cytoplasmic bridges, which was ultrastructurally demonstrated (Figure 2L,M and Figure 5Q,R), was evidenced using a fluorescent tracer. To this aim, 10 KDa rhodamine-labeled dextran was microinjected into a cell of spheroid and allowed to spread into neighboring ones.

The detected movement of the high-dimension tracking molecule among cells (Figure 8N–Q) validated the presence of these cytoplasmic canals and supported the possibility of large molecules passing from cell to cell, thereby providing a mutual exchange.

## 3. Discussion

MCF7 aggregation and complete formation of spheroids take place in about two weeks, owing to the synergistic and cooperative work of TNTs, amyloid fibrils, gap junctions, and large stable cytoplasmic bridges. These different processes, which appear mainly sequential, but also overlap to a lesser extent, correlate with important morpho-functional changes. Besides the events previously described in spheroid formation, such as intercellular adhesion [47,48,49], our results show that other phenomena concur in a 3D multicellular stable system, during both early (formation of TNTs and synthesis of amyloid fibrils) and late (opening of cytoplasmic bridges) steps of aggregation.

Owing to the aggregation and the opening of the intercellular communication systems, which also allow the flux of large molecules, each MCF7 cell takes advantage of the physical and chemical cooperative interactions among cells, enhancing its own fitness [50]. Moreover, this allows the coexistence, in the spheroid, of cells with very different phenotypes.

Overall, the 3D spheroid provides all the cells with a broader range of flexibility with respect to their own differentiation stage. Notably, ROS concentration has to be maintained within a certain range to reach the multicellular state [22]. Indeed, ROS is involved in the intercellular movements of molecules via the increase of TNT-biogenesis, the actin cytoskeletal reorganization, and by the formation of large cytoplasmic bridges. ROS production is also a prerequisite to protein folding in ER (i.e., amyloidogenesis) [28,29,36].

The multicellular state is achieved early by the formation of aggregates starting from the cell-to-cell approach carried out via the outgrowth of thin cytoplasmic projections. These structures are extended between neighboring cells and then remodeled in TNTs. The latter are highly dynamic and transient structures promoted by ROS [22] that connect various types of cancer cells [17,20,51] and also mediate communications in the first phases of MCF7 aggregation [20,51].

TNT structures can represent, also in spheroid formation, the first route to exchange resources, information and electrical signals [24] between distanced cells.

In early spheroid formation, TNTs were revealed in different ways: ultrastructurally and by immunolocalization of TEX14, one of the best characterized protein involved in the formation of cellular bridges [33,34,52,53], and of actin bundles longitudinally oriented with respect to TNTs’ major axis, and interpreted as a system to implement the structure stability.

Notably, in 24 h old aggregates, ROS production is also coupled to amyloidogenesis. The newly synthesized ThS-positive amyloid fibrils are made of Pmel17, one of the numerous proteins contributing to the formation of amyloid fibrillar structures in mammals [26,54]. Amyloid, owing to its intrinsic mechanical properties, maintains the cells in proximity and “glues” [27] them in a kind of resistant matrix.

The amyloidogenic deposition is regionally confined to account for the developmental phase; faced, but distanced cells show dilated cisternae of RER filled with this protein that is secreted among cellular interposed space. Later, in older spheroid, amyloid deposits are evident in the most superficial cells, where proliferation predominantly occurs, and where the cells are not in close reciprocal contact (i.e., amyloid material is exocytosed principally during the early phase of spheroid formation).

The detection of amyloid synthesis in the early phases of spheroid formation can be categorized as another example of a physiological function of amyloidogenesis (a process adopted from invertebrates to mammals, to package melanin and hormones or to transfer information in a wide range of organisms [26,27,28,29,55,56,57,58,59,60]).

The first type of cellular adhesion, made possible by the amyloid production, drives morphogenesis and cellular differentiation, resulting in cells with different phenotypes.

Together with the aforementioned functions, according to Rustom [19], both TNT biogenesis and amyloid fibril formation could be involved to keep ROS concentration in the physiological range, thus avoiding severe damages that may lead to apoptosis or necrosis [30]. As stress conditions resulting in the ROS build-up represent a well established starting point for carcinogenesis [61], we speculate that the procedures adopted for suspension culture of MCF7 drive them into a stress condition, resulting in ROS imbalance. Therefore, the protrusion of TNTs and amyloid deposition derive as a consequence and allow the aggregation and the relieving of the stress condition in the end.

Along with maturation, the connectivity among cells increases and the spheroids reach a complex organization owing to the massive genesis of specialized junctions, leading to a polarization of outer cells, which facilitates intercellular communications between inner cells.

MCF7 cells, when cultured in close contact, exploit another system to stick together and to reach direct communication between adjacent cells, by means of gap junctions. Indeed, the early presence of gap junctions among MCF7 spheroid-cells is evidenced here ultrastructurally and by immunolocalizations of typical junctional proteins such as E-cadherin and Connexin 43. Our data agree with a large body of previous experimental evidence [47,49,62] even if the increased or decreased expression of these markers in relation to potential development of malignancy in human breast is still debated.

The MCF7 cell-networking (in 11–16 days old spheroids) is reached owing to the opening of large intercellular bridges, an event well evident in the spheroids in the last phase of culturing, as demonstrated by the microinjection of 10 KDa dextran fluorescent molecule.

Once formed, these evolutionarily conserved intercellular communications [52,63,64] are fundamental in guaranteeing and establishing large stable canals by which large molecules and organelles can pass in a continuous flux.

This networking in large spheroid is useful for facilitating cell proliferation, to allow cell differentiation and their coordination during the following morphological changes, as well as to supply gap junction connections that are formed precociously but cannot function during the undifferentiated state typical of stem cells [65,66].

The stable expression of TEX14, required both during early mitosis (methaphase) [67] and for the formation of intercellular bridges [52,63], is localized in stem cells and in differentiating cells. In the cells forming mature spheroid, we link the expression of this factor to bridge formation rather than to cellular division phase, as mitotic events are numerically reduced and the number of cells connected by bridges increased.

During the different identified steps, multiple genes are concomitantly regulated; stemness markers (Sox2 and SSEA-4) are highly expressed in the earlier steps whereas they are expressed at a lower level in mature spheroids, where differentiated cells predominate. These changes in the expression level of several markers contribute to the variation in cell phenotype, starting from cells small in size with large nuclei, showing relaxed chromatin, up to larger differentiated cells with the cytoplasm filled with actin and intermediate filaments playing a scaffolding role. In particular, thickly bundled keratin deposits are generally considered as a parameter to assess the differentiation status of various types of cancer [68,69].

Synchronously and cooperatively, during the first step of spheroid growth, cells are characterized by production of ROS and overproduction of ACTH/α-MSH and IL18. By contrast, cell markers linked to the differentiation state such as E-cadherin, Connexin 43, and cytokeratins peaked starting from 5–7 days.

In the multi-layered mature spheroids (11–16 days), the cells, as already evidenced by many researchers, are grouped in an external aerobic zone represented by differentiated cells and an inner anaerobic one, where few cells with stem characteristics are placed and where they can stay in a niche protected and rich in nutrients [10,70].

The morphogenetic events, starting from the earlier steps, follow the general principle of cell–cell communication and cooperation to achieve a complex interconnected and multicellular unit [50].

Finally, the spheroid in its complexity (summarized in the schematic representation of Figure 9) has gained a condition of great benefit by reaching structural resistance to mechanical stress, as suggested by Ivascu and Kubbies [35], and the ability to regulate the composition of external/internal environments [71]. Spheroid maturation mimics the spatiotemporal sequence of morphogenetic steps occurring in embryogenesis/organogenesis, that is, MCF7 cells cultured in 3D are able to recapitulate the steps required for the in vivo formation of organs [72,73,74]. Thus, these strategies, starting from cell autonomy up to complex social interactions, and also present in non-tumorigenic MCF10A cell line, can be considered **as** an evolutionary conserved system [75].

Understanding the key events characterizing the spheroid maturation process may provide a useful platform for allowing therapeutic intervention. As generally accepted, even if each type of cancer requires a tailored treatment, the same type of tumor in different patients can respond differently to the same treatment. Thus, considering that the changes in a spheroid are so rapid and, in a few days, TNTs’ formation, production of amyloid fibrils, gap junction, and large cellular bridge formations can be detected, a number of questions arise: (1) Could an anti-cancer therapeutic approach fail owing to a mismatch between the selected therapy and one of the described developmental steps? (2) If the development of the spheroid is so rapid, would it be advantageous to proceed with simultaneous multiple attacks against amyloid synthesis, gap junction, and bridge formations? (3) Are antioxidant approaches, generically important in improving the cellular environment, a prerequisite for reducing or avoiding the aggregation phase? (4) Is the use of an important amyloid-degrading enzyme (NEP) an efficient method to clear the amyloid fibrils just produced?

In particular, the development of selected therapeutic approaches against specific targets such as amyloid fibril synthesis and stable large bridge opening (in both the initial and in the last phases of spheroid formation, respectively) may be of paramount importance as these two events are responsible for spheroid robustness and resistance to external attack.

On this premise, we show that, if the starting events are strictly linked to the redox homeostasis, which in turn regulate many functions such as amyloidogenesis, a way to slow down cellular aggregation could be to reduce the ability of cells to produce amyloid fibrils. We show that administration of NAC, which acts as a direct ROS scavenger [76,77,78], is a promising antioxidant procedure because the reduced synthesis of amyloid fibrils leads to smaller spheroids. We also hypothesize that another approach to reduce connectivity between cells could be obtained by the enzymatic action of NEP that degrades secreted amyloid fibrils, and consequently decreases the ability of the cells to aggregate by exploiting the intrinsic adhesiveness of amyloid fibrils.

## 4. Materials and Methods

All experiments were performed in five independent replicates.

All chemicals were purchased from Sigma-Aldrich (Saint Luis, MO, USA), unless otherwise indicated.

### 4.1. Cell Line and Monolayer Culture

The human breast adenocarcinoma MCF7 cells were obtained from the European Collection of Cell Cultures (ECACC, Porton Down, UK). Parental cells were cultured in 75 cm^2^ flasks in Dulbecco’s modified Eagle’s medium (DMEM), supplemented with 10% heat-inactivated fetal bovine serum (FBS), 1% L-glutamine, and 1% penicillin/streptomycin (all from Euroclone, Milan, Italy) at 37 °C under a 5% humidified CO_2_ atmosphere. All experiments were performed on cultures that were 80% confluent.

### 4.2. Aggregate and Spheroid Culture

The monolayer cells were harvested and dissociated into single cell suspension by trypsinization. Aggregate and spheroid formation was performed by plating the cells at a density of 5 x 10^4^ cells into ultra-low attachment six-well plates (Corning, Turin, Italy), in 2 mL/well of a serum-free medium consisting of DMEM/F12 medium, supplemented with 0.025% human epidermal growth factor (hEGF, R&D Systems, Minneapolis, MN, USA), 2% B27 (Gibco, Thermo Fisher Scientific, Waltham, MA, USA), 0.05% insulin, 0.0034% hydrocortisone, and 0.05% heparin.

Spheroids’ formation was monitored daily and, every 3 days, 300 µL of new medium with fresh factors was added.

### 4.3. Morphological Analysis

Aggregates and spheroids cultured in 3D suspension conditions were collected and analyzed at different indicative time points: 24 h; 3 days; 5–7 days; and 11–16 days.

Each step of spheroid manipulation was accompanied by a short centrifugation at 1000 rpm for 5 min.

#### 4.3.1. Measures of Aggregates and Spheroids

Aggregates and spheroids were imaged by phase contrast microscopy and measured according to Kelm and co-workers [79]. The mean geometric diameter (d) of the cultured spheroids was calculated using the following equation: d = (a × b)^1/2^, where a and b are orthogonal diameters of the spheroid. The average size of the cultured spheroids was reported as mean diameter ± SEM.

#### 4.3.2. Light Microscopy, and Transmission (TEM) and Scanning (SEM) Electron Microscopy

Samples were collected and fixed with 4% glutaraldehyde in 0.1 M Na-cacodylate buffer (pH 7.4) and then washed three times for 10 min in the same buffer.

For TEM analysis, specimens were post-fixed for 20 min with 1% osmic acid in cacodylate buffer. After standard dehydration in an ethanol series, samples were embedded in an Epon-Araldite 812 mixture and sectioned with a Reichert Ultracut S ultratome (Leica, Nussloch, Germany). Semithin sections were stained by conventional methods (crystal violet and basic fuchsin) and were observed with a light microscope (Eclipse Nikon, Amsterdam, Netherlands); images were acquired with a Nikon DS-SM camera. Thin sections were stained by uranyl acetate and lead citrate and observed with a Jeol 1010 electron microscope (Jeol, Tokyo, Japan).

For SEM analysis, specimens were post-fixed for 1 h in a solution of 1% osmium tetroxide and 1.25% potassium ferrous-cyanide and dehydrated (ethanol series and hexamethyldisilazane). The samples, mounted on carbonated stubs and gold coated in an Emitech K250 sputter coater (Emitech, Baltimore, MD, USA), were observed with a SEM-FEG XL-30 microscope (Philips, Eindhoven, Netherlands).

#### 4.3.3. Immunocytochemistry

To verify the status of stem cells, the expression of a surface marker (SSEA-4) and a transcription factor (Sox2) was evaluated [6,37,38].

The presence of cell differentiation markers was visualized; Connexin 43 and E-cadherin, involved in gap junction formation [35,47,48,49], cytokeratin (for bundles) [41], and TEX14 (essential component of the intercellular bridges) [52].

Amyloid fibril characterization: amyloid structures were identified according to Le Vine III [80] by staining with ThS and visualizing the amyloid-specific green/yellow fluorescence with a Leica laser confocal microscope. Amyloid fibrils were also localized immunocytochemically, using an antibody directed against Pmel17, which is one of the proteins able to form amyloid fibrillar structures in mammals [26,54,81,82] (Table 1).

The presence of ACTH and its cleavage product (owing to degrading enzyme NEP) α-MSH, and the proinflammatory cytokine IL18, as generally linked to amyloidogenesis [39], was evaluated.

Briefly, cells were fixed with 4% paraformaldehyde in PBS (pH 7.4) and permeabilized with 0.1% Triton X-100 in PBS. After 1 h incubation with a blocking solution (2% bovine serum albumin, 0.1% Triton X-100 in PBS), they were incubated overnight at 4 °C with the primary antibodies (Table 1). After three washes of 10 min, incubations with suitable secondary antibodies, conjugated with cyanin 5 (Cy5, Abcam, dilution 1:200), were carried out for 1 h at room temperature. Nuclei were counterstained with 4′,6′-diamino-2-phenylindole (DAPI).

In control samples, primary antibodies were omitted.

#### 4.3.4. Actin Filaments

F-actin filaments (cell differentiation marker) were identified by incubating cells with rhodamine-labeled phalloidin (50 µg/mL in PBS) for 1 h. After extensive washes, coverslips were mounted with Citifluor and slides were observed either under an Eclipse Nikon microscope or a Leica laser confocal microscope.

#### 4.3.5. Tunneling Nanotubes and Intercellular Bridge Characterization

TNTs and intercellular bridges were localized immunocytochemically using an antibody directed against TEX14 as an essential component [33,34,52,53] (Table 1).

### 4.4. Western Blot

Lysates from aggregates and spheroids were obtained using RIPA buffer (50 mM Tris-HCl pH 7.5, 50 mM NaCl, 0.1% SDS, 1% NP40, 0.5% sodium deoxycholate, protease/phosphatase inhibitors cocktail). The lysates were clarified by centrifugation and protein concentration was assessed by Bradford assay (Serva, Heidelberg, Germany). Protein extracts were subjected to 8 or 10% SDS-PAGE (40 or 60 µg protein each lane) and then transferred onto 0.45 μm pore size nitrocellulose membranes (Amersham Protran Premium, GE Healthcare, Chicago, IL, USA). The filters were blocked for 2 h at room temperature with 5% (*w*/*v*) non-fat dried milk in TBST (Tris-buffered saline containing 0.1% Tween-20) and then incubated for 2 h at room temperature (r.t.) with the following primary antibodies (diluted in TBST/5% milk, Table 1): Sox2, Connexin 43, Tex14, Pmel17, and IL18; detection of glyceraldehyde 3-phosphate dehydrogenase (GAPDH) was used as loading control. After three washes of 10 min in TBST, the membranes were incubated for 1 h with horseradish-peroxidase conjugated anti-rabbit (dilution 1:7500; Jackson ImmunoResearch Laboratories, West Grove, PA, USA) or anti-mouse (dilution 1:4000; Jackson ImmunoResearch Laboratories) secondary antibodies. The membranes were exposed to the enhanced chemiluminescence substrate (LiteAblot PLUS, Euroclone), followed by autoradiography on X-ray film (KODAK Medical X-Ray film, Z&Z Medical, Iowa, USA). Densitometric analysis was performed with ImageJ software. The values are reported as relative optical density of the bands, normalized to GAPDH.

### 4.5. ELISA Assay

The human IL18 ELISA kit (MBL, Woburn, MA, USA) quantitatively measures human IL18 by sandwich ELISA. According to the manufacturer’s protocol, fresh samples (cell culture supernatants, pure or diluted) are incubated in the microwells coated with an anti-human IL18 monoclonal antibody. After washing, the peroxidase conjugated anti-human IL18 monoclonal antibody is added. After another washing, a mixture of the substrate reagent with the chromogen was added to the plate. Finally, an acidic solution terminates the enzyme reaction and stabilizes the colour development. The optical density (OD) of each microwell is measured at 450 nm using a microplate reader (Tecan, ThermoFisher Scientific). The concentration of human IL18 (expressed in pg/mL) is calibrated from a dose-response curve based on the reference standards.

### 4.6. Functional Studies

#### 4.6.1. Intracellular ROS Evaluation

The evaluation of ROS level during the first steps of cell aggregation (characterized by protrusion of TNTs and amyloid fibril production) is of paramount importance [22,28,29,31,82,83].

The 2′,7′-dichlorodihydrofluorescein diacetate (H2DCFDA Molecular Probes), a fluorigenic probe commonly used to detect the overall degree of intracellular level of ROS, was used to evaluate ROS production.

H2DCFDA is a non-fluorescent compound that readily crosses cell membranes. It is hydrolyzed to 2′,7′-dichlorofluorescein (DCF) within cells and becomes fluorescent when it is oxidized by ROS. Oxidation can be detected by monitoring the increase in fluorescence. As previously described [83], cells were incubated in the dark with 10 mM H2DCFDA for 30 min at 37 °C and then washed three times for 10 min with HBSS buffer. Fluorescence was determined by excitation at 488 nm and emission at 525 nm wavelength; fluorescent images were captured with a Leica laser confocal microscope.

#### 4.6.2. Treatment with ROS Scavenger

NAC is an antioxidant that is generally utilized to mitigate oxidative stress condition owing to increased ROS in the cell [76,77].

MCF7 cells were seeded onto ultra-low attachment plates, as previously described, and instantly treated with 10 mM NAC. Floating 3D cellular structures were observed after 24 h and 3 days, in order to evaluate the status of aggregation, compared with untreated samples. Images were acquired (phase contrast) utilizing an Olympus IX51 microscope.

#### 4.6.3. Treatments with NEP Degrading Enzyme

Neutral endopeptidase/Neprilysin/CD10 (NEP; a 93-kDa zinc metallo-endopeptidase) is an important amyloid degrading enzyme. This enzymatic degradation plays an integral role in the clearance of amyloidogenic products and has been shown to be highly critical for removing amyloid peptides in Alzheimer’s disease [46]. MCF7 cells were seeded onto ultra-low attachment plates and treated with 5 ng/mL NEP; the formation of aggregates was observed after 24 h and 3 days. Images were acquired (phase contrast) utilizing an Olympus IX51 microscope.

#### 4.6.4. Fluorescent Dextran Microinjection

Functional trafficking activity via intercellular bridges was monitored using 10 KDa rhodamine-labeled dextran (Molecular Probes). According to Pennarossa and coworkers [63], 50 mg/mL of the tracking molecule was directly microinjected into the cytoplasm of a single cell of a mature spheroid (up to 11 days), with an Eppendorf FemtoJet^®^ Microinjector (Eppendorf) using 100 hPa injection pressure for 0.3 s; compensation pressure was set to 40 hPa. The temperature was maintained at 37 °C during the experimental acquisitions. The movement of fluorescent dextran was continuously monitored, for about an hour, under an Eclipse TE200 inverted microscope (Nikon), equipped with Digital Sight DS-2MBW Camera (Nikon). The most significant pictures were acquired and processed using NIS-Elements Advanced Research (AR) software (Nikon, Tokyo, Japan).

#### 4.6.5. Statistical Analysis

Data are presented as means (±SEM) of five independent replicates. Statistical differences were calculated by one-way ANOVA followed by Tukey’s post-hoc test; different letters indicate statistically significant differences (*p* < 0.01).

## 5. Conclusions

This study offers an opportunity to better understand the complexity of MCF7 spheroid formation. Ongoing analysis will lead to increased elucidations of the mechanisms by which tumor cell interactions take place.

Identification of these parameters may be a step towards the development of a more specific and even personalized therapy.

## Figures and Tables

**Figure 1 ijms-21-05400-f001:**
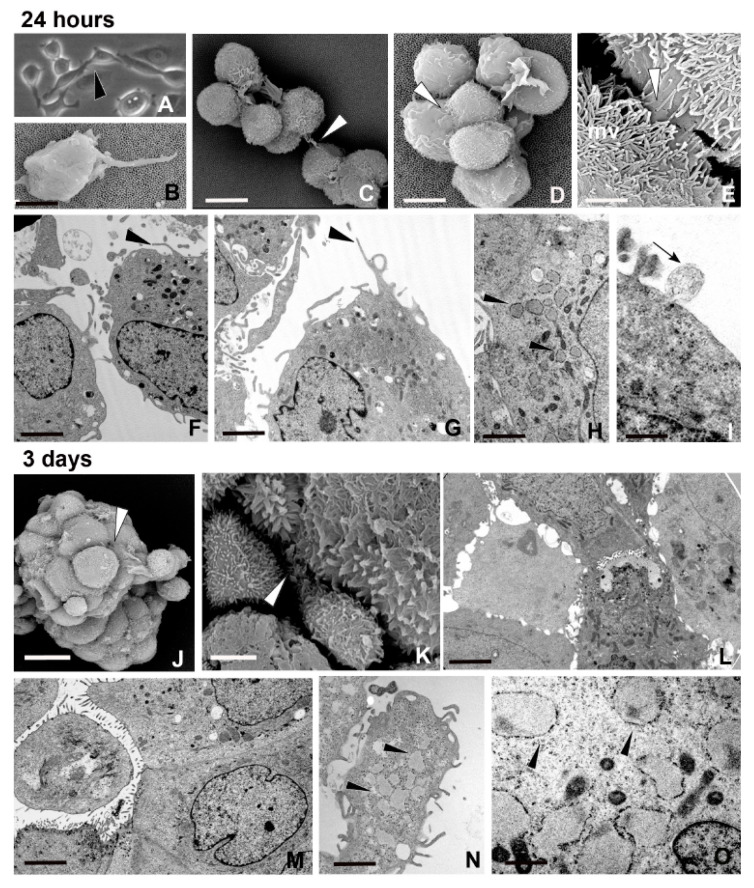
Morphology and behavior of cells during spheroid formation. (**A**–**I**) Twenty-four hour spheroid culture. Phase-contrast micrograph of free-floating and of tiny aggregated human breast adenocarcinoma cells (MCF7) cells (**A**) showing projections among cells (arrowhead) and ultrastructural (scanning electron microscopy, SEM) detail (**B**) of tunneling nanotubes (TNTs) protruding from a single cell. Scale bar: 4.5 μm. (**C**–**E**) SEM micrographs. The dimension of the aggregates is variable and the spatial organization of MCF7 cells is based on differing cell interactions. In smaller aggregate, few cells characterized by uniform dimension are in proximity (**C**) (white arrowhead), while in compacted ones, most of the cells are in close proximity or in contact (**D**) (white arrowhead), interacting (**E**) by short microvilli (mv) embedded in the dense amyloid matrix (white arrowhead). Scale bars: (**C**) 9 μm; (**D**) 9.5 μm; (**E**) 2 μm. (**F**,**G**) Transmission electron microscopy (TEM) analysis. MCF7 cell surfaces are characterized by microvilli and by TNT-like structures (arrowheads) reaching the neighboring cells. Scale bars: (**F**) 1.2 μm; (**G**) 2 μm. (**H**,**I**) TEM analysis. Cells are characterized by large dilated reticulum cisternae filled with spatially organized fibrillar material (**H**) (arrowheads) and released exosomes (**I**) (arrow). Scale bars: (**H**) 0.5 μm; (**I**) 0.6 μm. (**J**–**O**) Three day spheroid culture. (**J**,**K**) SEM analysis. MCF7 cells are assembled in small round spheroid uniform in size. Cells are in close proximity or in close contact and show different shape and size (arrowheads). Scale bars: (**J**) 15 μm; (**K**) 3.3 μm. (**L**–**O**) TEM analysis. Cells, polygonal in shape, are discontinuously connected (**L**–**M**), while they are distanced at the external side of the spheroid (**N**). These peripheral cells show numerous dilated rough endoplasmic reticulum (RER) cisternae filled with fibrillar material within the cytoplasm (**N**,**O**) (arrowheads). Scale bars: (**L**) 2.5 μm; (**M**) 4 μm; (**N**) 1.7 μm; (**O**) 0.6 μm.

**Figure 2 ijms-21-05400-f002:**
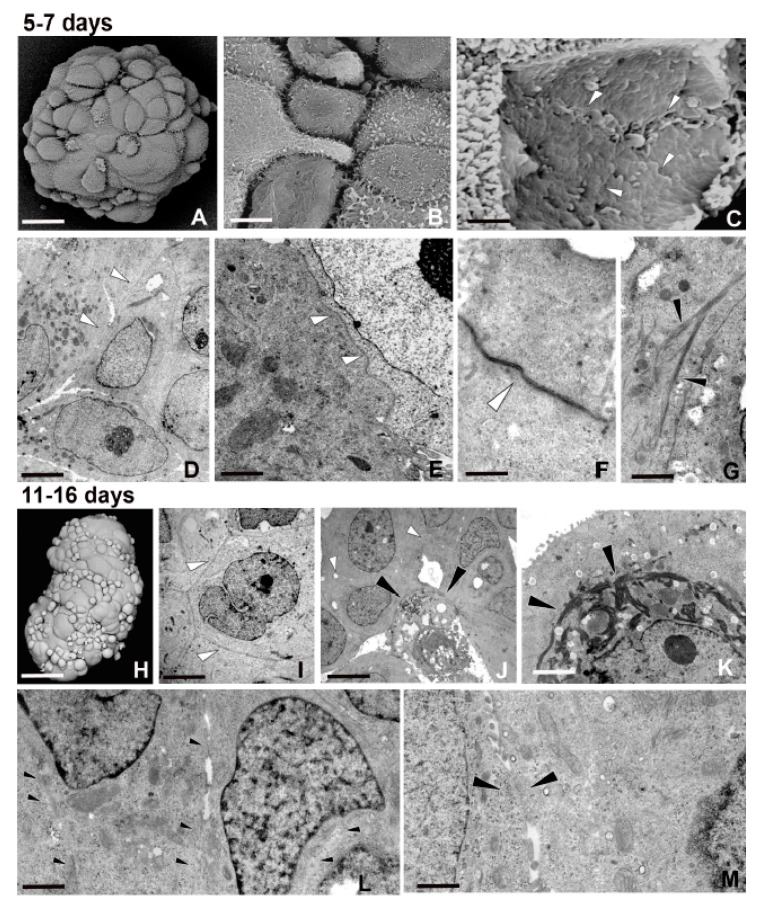
Morphology and behavior of cells during spheroid formation. (**A**–**G**) Five–seven day spheroid culture. (**A**–**C**) SEM micrographs of round growing spheroids that grow. Cells are different in shape and size as evident by the enlargement (**B**). The gap junctions (**C**) (arrowheads) among the surfaces of the inner cells are evident after removal of some peripheral ones. Scale bars: (**A**) 30 μm; (**B**) 5 μm; (**C**) 0.7 μm. (**D**–**G**) TEM analysis. (**D**) Note the high nucleus–cytoplasm ratio. The presence of euchromatic nucleus with voluminous nucleoli reflects an intense metabolic activity. The cells are in close contact (**D**–**E**) (arrowheads). In the detail (**F**), the unambiguous gap junction profile is shown (arrowhead). (**G**) Many cells are characterized by bundles of actin filaments localized under the cytoplasmic membrane (black arrowheads). Scale bars: (**D**) 3.8 μm; (**E**) 1.7 μm; (**F**) 1 μm; (**G**) 2 μm. (**H**–**M**) Eleven–sixteen day spheroid culture. SEM analysis (**H**). Large multicellular spheroids are made of tightly packed cells, different in shape and size, showing various differentiation states. Scale bar: (**H**) 50 μm. (**I**–**M**) TEM analysis. Cells are in close contact (**I**,**J**) and in the inner area of the spheroid, apoptotic cells are evident (**J**) (arrowheads). The presence of thick dark bundles of filaments (arrowheads) surrounding the nucleus suggests an ongoing cell differentiation (**K**). The large pattern of cell-to-cell interconnections (**L**,**M**) is owing to the opening of large stable cytoplasmic bridges (**M**) (arrowheads). Scale bars: (**I**) 4.2 μm; (**J**) 8 μm; (**K**) 1.25 μm; (**L**) 1.7 μm; (**M**) 0.7 μm.

**Figure 3 ijms-21-05400-f003:**
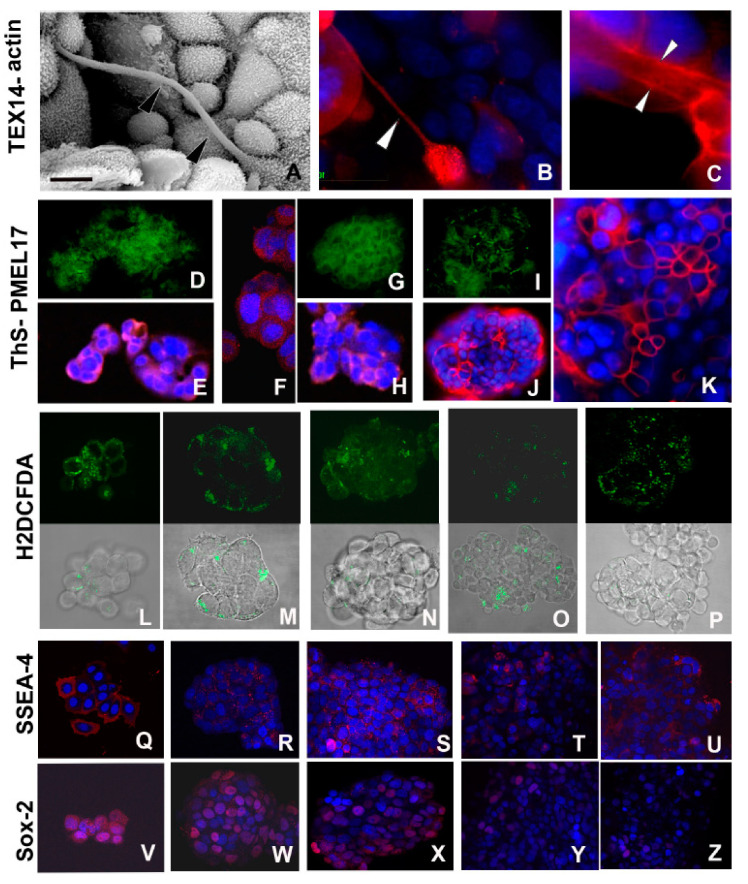
Morpho-functional characterization of cells during early spheroid formation. (**A**–**C**) SEM (**A**) and immunofluorescence (**B**,**C**) analysis of MCF7 cells during the early phases of aggregation. Cell communications are established via TNTs, evidenced ultrastructurally (**A**) (arrowheads) and by immunofluorescence stainings (**B**,**C**). The nanocanals are confirmed by intensive labelling showing the expression of testis expressed gene 14 (TEX14) (**B**) (arrowhead) and of actin (**C**) confined along the internal face of TNT plasma membrane (arrowheads). Nuclei are stained with 4′,6′-diamino-2-phenylindole (DAPI) and marked in brilliant blue. Scale bar: (**A**) 7 μm. (**D**–**K**) Identification of amyloid fibrils localized by bright green fluorescence of thioflavine S (ThS) and by expression of melanocyte protein (Pmel17) (red) in MCF7 cells during early aggregation phase (24h) (**D**,**E**), after 3 days (**F**,**H**), and only superficially in mature spheroids at 5–7 days (**I**–**K**). Nuclei are stained with DAPI. Separate staining for ThS (**D**,**G**,**I**) and Pmel17 (**E**,**F**,**H**,**J**,**K**), are proposed to better evaluate each signal identifying the presence of amyloid materials. (**L**–**P**) The overall degree of cytoplasmic ROS is detected using the fluorescence dye 2′,7′-dichlorodihydrofluorescein diacetate (H2DCFDA). Comparison among the upper series of photographs showing only the staining for ROS with the related lower series, showing superimposed bright field images and ROS localization, is proposed to provide a better evaluation of the signal strength and its localization in the various developmental stages of spheroid. The signal (green) persists during all developmental stages of spheroid with not-significative decrease in the last phase of development. Early (24 h) aggregation phase (**L**,**M**), 3 days spheroid (**N**), 5–7 days (**O**), and 11–16 days spheroid culture (**P**). (**Q**–**Z**) Comparison of the expression of stemness markers stage-specific embryonic antigen-4 (SSEA-4) (**Q**–**U**) and SRY (sex determining region Y)-box 2 (Sox2) (**V**–**Z**) from 24 h aggregates (**Q**,**V**), and 3 (**R**,**W**), 5–7 (**S**,**X**), and 11–16 (**T**,**Y**,**U**,**Z**) days spheroids. The signal decreases along with maturation of spheroids and it is superficially confined, as evidenced in the overview of selected peripheral cells of spheroid 11–16 days old (**U**,**Z**)

**Figure 4 ijms-21-05400-f004:**
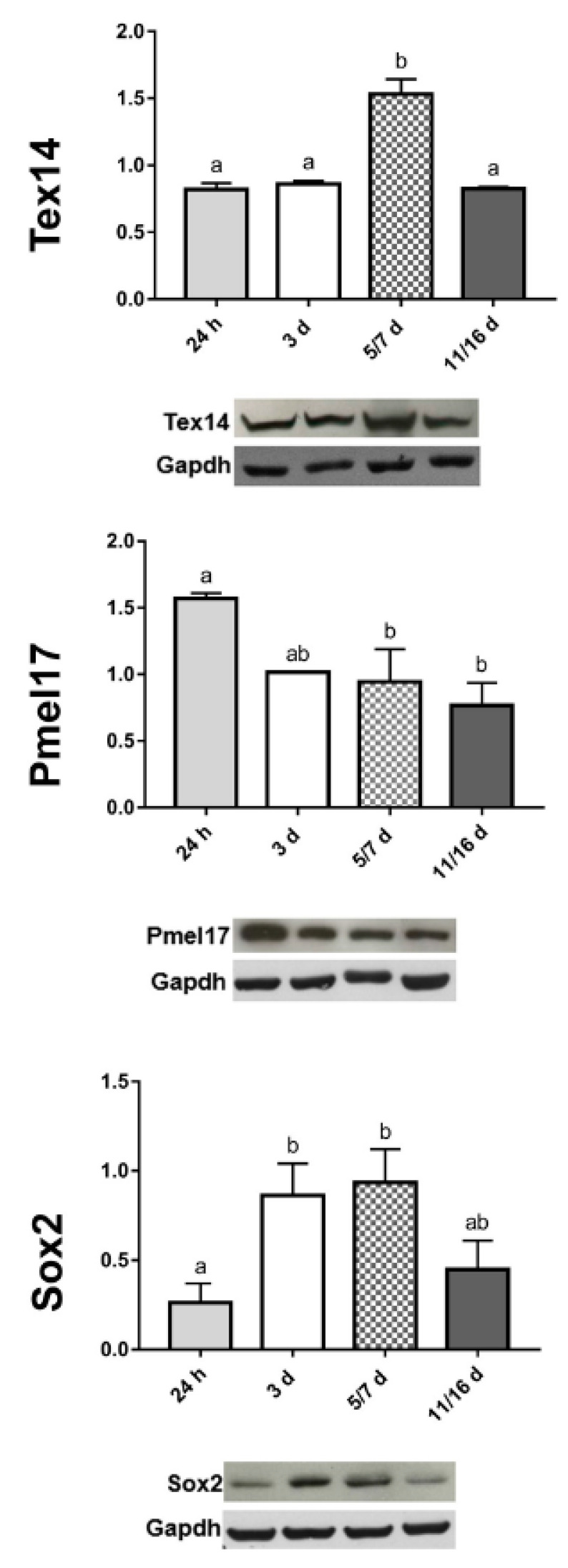
Semiquantitative/quantitative analysis on spheroids. The graphs illustrate the expression levels of TEX14-intercellular bridge forming factor, the Pmel17 (amyloidogenic protein), and the Sox2-stemness marker, respectively, at different stages of spheroid formation. The data result from a densitometric analysis of the Western blots. The values are reported as relative optical density of the bands normalized to glyceraldehyde 3-phosphate dehydrogenase (GAPDH). Statistical differences were calculated by one-way ANOVA followed by Tukey’s post-hoc test; error bars represent SEM and different superscripts denote statistically significant differences (*p* < 0.01).

**Figure 5 ijms-21-05400-f005:**
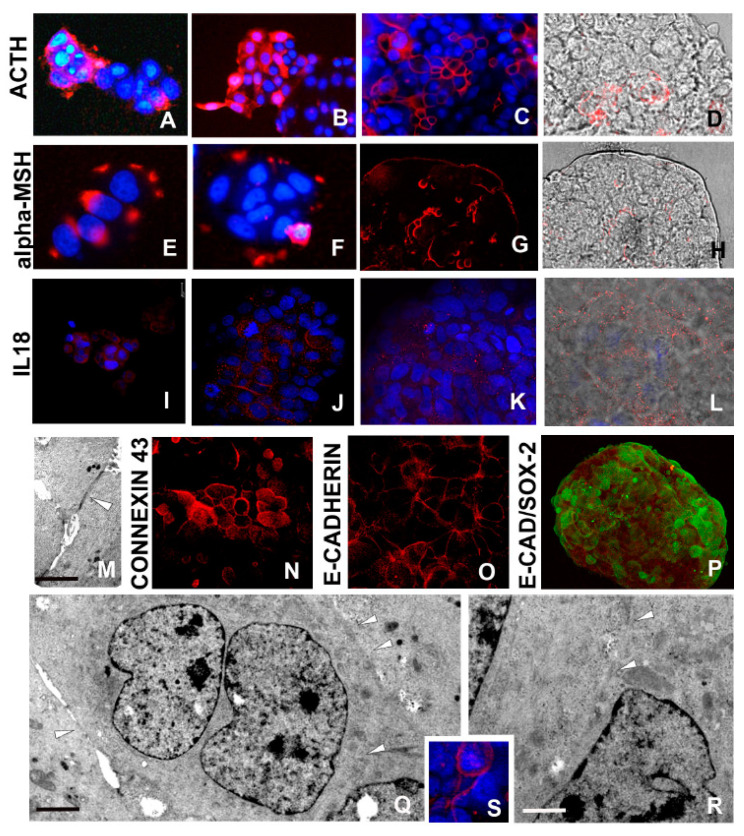
Morpho-functional characterization of cells in developed spheroids: ACTH/α-MSH, interleukin (IL)18 expressions and opening of stable intercellular bridges (**A**–**H**). Laser confocal microscope analysis. Representative micrographs depicting ACTH and α-MSH expression by immunocytochemical characterization. At 24 h (**A**,**E**) and 3 days (**B**,**F**) spheroid culture, the signal is quite strong, whereas in mature spheroids (**C**,**D**,**G**,**H**), the signal is superficially localized. In D and H panels, red signal (immunolocalization) and bright field were superimposed to better identify the involved area of spheroid. (**I**–**L**) IL18 expression is localized in most cells in early aggregates (**I**,**J**) and especially in peripheral cells of mature spheroid (**K**,**L**). In L panel, red signal (immunolocalization) and bright field were superimposed. (**M**) TEM analysis. Two neighboring cells are in close contact according to the presence of specialized junctions (arrowhead). Scale bar: (**M**) 2 μm. (**N**–**P**) Confocal microscopy images. Starting from 5–7 days, cells are connected by gap junctions that are characterized by Connexin 43 (**N**) and E-cadherin (**O**) expression. Staining with antibody raised against E-cadherin (**O**) shows a punctate pattern at the periphery of the cells. (**P**) Confocal microscopy images. Double labelling of spheroid cells with antibodies raised against E-cadherin (red) and Sox2 (green) showing that stemness (Sox2) does not match with the gap formation that is considered critical event in growth and differentiation. (**Q**,**R**) TEM analysis of MCF7 cells showing the presence of intercellular bridges that allow cytoplasmic continuity (arrowheads). Scale bars: (**Q**) 2.5 μm: (**R**) 2 μm. (**S**) Immunofluorescence staining showing expression of TEX14 (red signal) lining the MCF7 cells. Nuclei are stained with DAPI.

**Figure 6 ijms-21-05400-f006:**
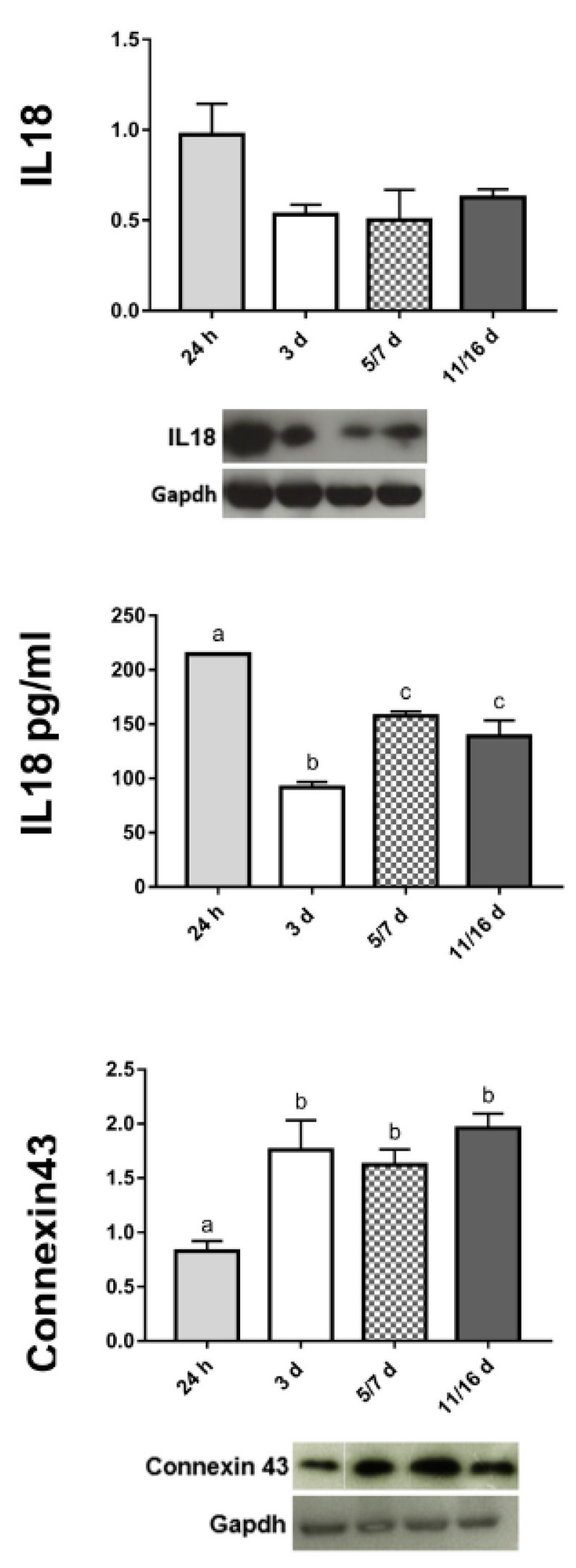
Semiquantitative/quantitative marker analysis on spheroids. The graphs illustrate the expression level of IL18-amyloidogenic associated cytokine; the IL18 secretion in the medium evaluated by ELISA assay; the expression level of Connexin 43-gap-junction’s component. The expression levels, at different stages of spheroids’ formation, resulted from a densitometric analysis of the Western blots. The values are reported as relative optical density of the bands normalized to GAPDH. Statistical differences were calculated by one-way ANOVA followed by Tukey’s post-hoc test; error bars represent SEM and different superscripts denote statistically significant differences (*p* < 0.01).

**Figure 7 ijms-21-05400-f007:**
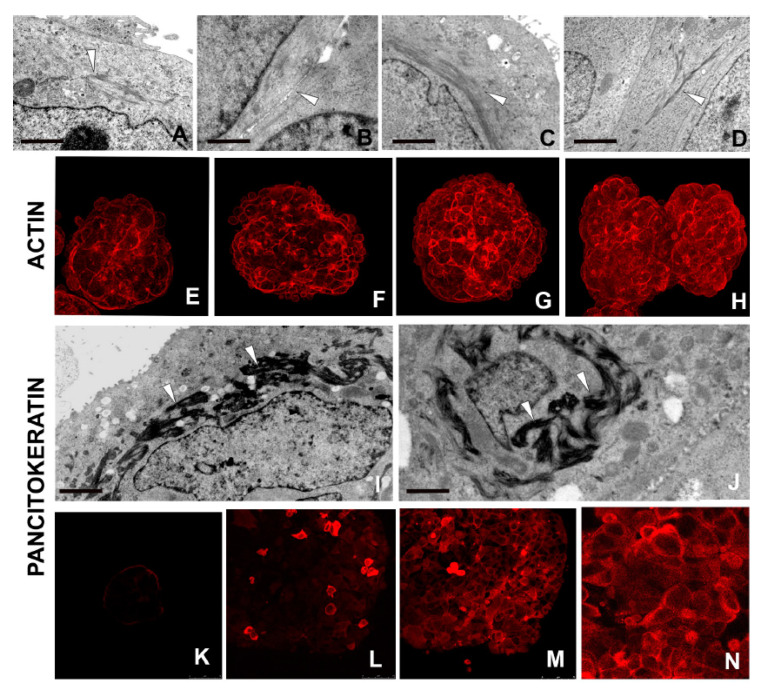
Morpho-functional characterization of cells in developed spheroids. (**A**–**H**) TEM and immunocytochemical analysis. In well-developed spheroids, the MCF7 cells show differentiation markers such as actin filaments forming bundles located close to the membrane or surrounding the nucleus (**A**–**D**) (arrowheads). Immunofluorescence staining (**E**–**H**) validate the morphological analysis by showing the highly expressed actin protein (red signal) in well-developed spheroids. Scale bars: (**A**) 1 μm; (**B**) 1 μm, (**C**) 1.7 μm, (**D**) 2 μm. (**I**–**N**) TEM and immunocytochemical analysis. Ultrastructural analyses reveal the characteristic keratin bundles (**I**,**J**) (arrowheads) also visualized using an antibody against pan-cytokeratin (**K**–**N**). Cytokeratins are present in the MCF7 differentiated cells forming spheroid at 3 days (**K**), 5–7 days (**L**), and 11–16 days (**M**) spheroids. Detail (**N**) of panel M. Scale bars: (**I**) 1.6 μm; (**J**) 1.6 μm.

**Figure 8 ijms-21-05400-f008:**
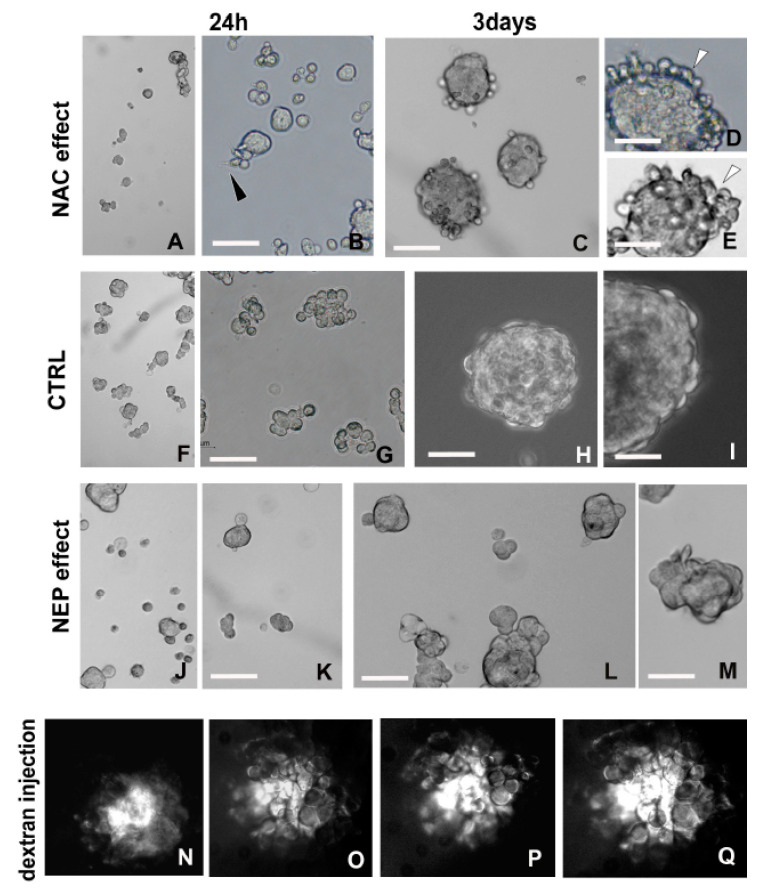
Functional analysis. (**A**–**I**) Comparison between MCF7 cells after 24 h and 3 days with (**A**–**E**) and without (**F**,**I**) NAC treatment. The antioxidant action of this molecule, acting on amyloidogenesis, reduces the MCF7 ability to form spheroids, whose size was smaller and variable in size compared with controls already at 24 h post treatment. (**G**–**I**). Cells of early aggregates (**B**) are small in size and show protruding thin expansion (arrowhead). Scale bars: (**B**) 45 μm; (**C**) 50 μm; (**D**,**E**) 30 μm; (**G**) 38 μm; (**H**) 45 μm; (**I**) 24 μm. (**F**–**M**) Comparison between MCF7 cells after 24 h and 3 days without (**F**–**I**) or with NEP treatment (**J**–**M**). From 24 h up to 3 days, the enzymatic degradation by NEP treatment, triggering a removal of amyloid fibrils involved in the adhesion of cells, leads to very small spheroids characterized by an irregular profile. Scale bars: (**K**–**M**) 32 μm. (**N**–**Q**) Fluorescent dextran is microinjected in the cytoplasm of a single cell. Starting from about 5 min, the tracking molecule flows into the adjacent cells, evidencing the intercellular connections and demonstrating the functional activity of these cytoplasmic canals.

**Figure 9 ijms-21-05400-f009:**
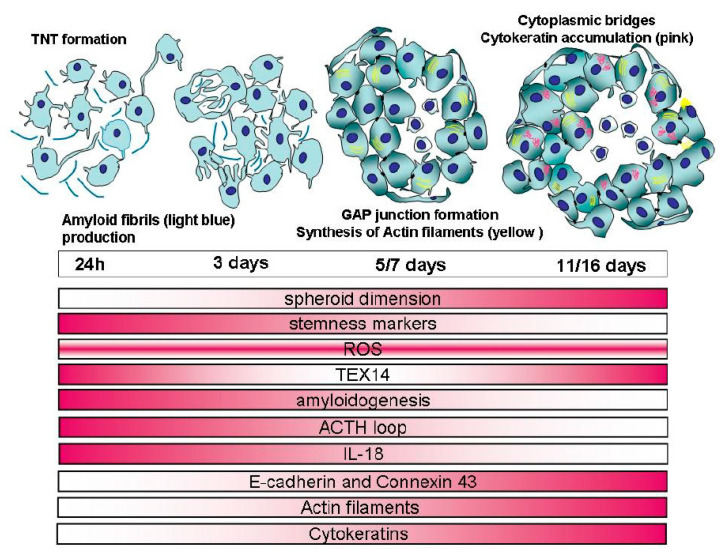
Graphical view of spheroid formation with the principal morphological evidences (TNTs, amyloid fibrils, gap junctions, actin filaments bundles, and cytokeratins accumulations). In the lower panel, the repertoire of the specific cellular events is represented with red color code, with color gradients corresponding to the time points.

**Table 1 ijms-21-05400-t001:** List of primary antibodies used for immunocytochemical and immunoblotting studies. Stemness markers: SSEA-4, stage-specific embryonic antigen-4; Sox2, SRY (sex determining region Y)-box 2; TEX14, testis expressed gene 14; Pmel17, melanocyte protein; IL-18, interleukin-18; ACTH, adrenocorticotropin hormone; α-MSH, alpha melanocyte-stimulating hormone; GAPDH, glyceraldehyde 3-phosphate dehydrogenase.

Antibody	Description	Company	Application	Dilution
SSEA-4	Mouse monoclonal [MC813]	Cell Signaling Technology	IF	1:500
Sox2	Rabbit polyclonal	Proteintech	IF	1:100
			WB	1:2000
Connexin 43	Rabbit polyclonal	Assay biotech	IF	1:100
			WB	1:500
E-cadherin	Mouse monoclonal [SHE78-7]	Calbiochelm	IF	1:500
Pan-Cytokeratin	Mouse monoclonal [C-11]	GeneTex	IF	1:200
TEX14	Rabbit polyclonal	Proteintech	IF	1:200
			WB	1:1000
Pmel17	Mouse monoclonal [HMB45]	Abcam	IF	1:200
			WB	1:400
ACTH	Rabbit polyclonal	Abcam	IF	1:100
α-MSH	Rabbit polyclonal	Abcam	IF	1:100
IL18	Rabbit polyclonal	Thermo Scientific	IF	1:50
			WB	1:1000
GAPDH	Rabbit polyclonal	Proteintech	WB	1:8000

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
