# Peer review of "MCF7 Spheroid Development: New Insight about Spatio/Temporal Arrangements of TNTs, Amyloid Fibrils, Cell Connections, and Cellular Bridges"

_ijms, 2020, doi:10.3390/ijms21155400_

Round 1

Reviewer 1 Report

General comments

This is an interesting study and the authors have presented some valuable new data in relation to MCF7 spheroids, that beside being a largely used model for drug screening, still lacks a deeper characterization.

The manuscript presented a good range of techniques, however, in my opinion, some improvements must be performed.

First, the English needs to be reviewed, some sentences are written in a way that make it difficult for the reader.

The introduction could be improved, providing more information in relation to the studied characteristics in MCF7 spheroids, providing a broader view of the state of the art in relation to what is known about the role of the TNTs and amyloid fibrils; gap junctions; cytoplasmic bridges in spheroids aggregation, or of there is no available information highlight in the text.

The methods section lacks much important information necessary for replicating the used methodology. Many theorical descriptions in the methods, some should be in the introduction.

The results should be rewritten to provide only the description of the obtained results, removing comments and references related to literature information that should transferred to other sections, according to what is more reasonable.  Alternatively, the guidelines for the author of this journal allow to combine the discussion with the results, but in this case, it would be a Results/Discussion section. Also, most TEM presented images are from low magnifications, which does not provided support for the detailed descriptions, better images with higher magnifications should be used.

There are other points along the manuscript that require improvements. Below I have provided some suggestions and some requirements to provide more evidence and quality for you work.

Given the above-mentioned considerations the manuscript requires major revisions.

ABSTRACT

The authors should uniformize the verb tense (past) when reporting the obtained results.

Line 22: the techniques used in this manuscript haven’t studied the genes expression, otherwise their respective proteins.

INTRODUCTION

Some sentences from the introduction are clumsily constructed, making it not clear for the reader at first reading. I suggest you, not to use so long sentences Eg. Lines 54-56. Lines 65-68. Try to divide it into two sentences.

Line 60: please insert a reference for this sentence.

Line 61 and 62: Please provide a reference supporting if this is described in another or in this cell line spheroids.  The reference in the end of the paragraph on support the second sentence, the presence of cytoplasmic bridges or sharing cytoplasm is not descripted in the reference provided.

Line 67: I believe the verb is sheds

I felt that there is not enough information in the introduction about the topics studied, TNTs and role of amyloid fibrils, cytoplasmic bridges. Is there any information in the literature in relation to MCF7 or other breast cell lines spheroids?

Also why is important study the presence of stem cells in the spheroids? Why is important to study ROS in the amyloid fibrils formation?

RESULTS

General comments:

I would like to congratulate the authors for the great pictures of SEM, it was very nice to the MCF7 spheroids from that perspective.

In my opinion, there are a high number of references in the Results section. Normally, this section is more straightforward, just reporting concisely the results with the used techniques. The references related to the techniques should be included in the other sections, mainly the discussion section. However, some can be included in the methods to justify why that methods are being applied. The instructions to the authors refer that discussion may be combined with the results, but in this case, there would be no separate section.

Line 75: variously differentiated: is too vague. The authors could find a more precise way to described it.

Lines 84 and 85: RER cisternae filled with fibrillar material. This description is not visible in the presented the images, the magnification is not sufficient to evidence the described detail.

Lines 98 and 99: Use the same formatting used in these lines for 3 days spheroid culture, for highlighting the 24 hours in culture, use underline format.

Lines 84 and 85, 109: The described fibrillary pattern of the content inside RER cisternae is not evident in the image magnification. If the authors want to prove the fibrillary pattern higher magnification should be used, at least include an insert in the presented image of high magnification.

Lines 132 and 133: Please uniform the spaces between the number and the unit, some are with space and other without it. E.g 30µm and 0.7 µm, I would prefer with space, it makes it easier for reading. The same occurred with the space after the full stops. Please review all the manuscript to find other similar situations.

Line 133: In my opinion theoretical information such as the nucleus-cytoplasm ratio is high hallmark of stem cells, should not be included in captions. It is relevant information, but it is more appropriate in the discussion section.

Line 138: Scale bar from images E and F: It seems to me that the scale bars of E and F should not be the same. F seem to be a higher magnification. Please check this question.

Line 142: It would benefit to present a higher magnification of the filament bundles, ate least in an insert in the same image.

Line 143: To evidence the cellular bridges, higher magnification should be used. Please describe better what you consider cellular bridges.

Lines 162 to 165: This information belongs to the discussion, from my point of view.

Line 165: how can you affirm: “having a short life span”, is it a literature information? If yes, please insert the reference. If not, how did you evaluated the life span of the TNTs.

Lines 203 to 212: In my opinion this information is more appropriate to the discussion section.

Lines 213: In order to prove the ultrastructural fibrillar aspect, better images, with higher magnifications, must be presented.

Line 230: Please insert a reference. Also,I believe this sentence is more appropriate to the introduction or discussion.

Lines 231 to 233: Also, I believe this sentence is more appropriate to the introduction or discussion.

Line 245: This sentence does not give any information to the readers; please review what idea you would like to let the readers know.

Lines 247 and 248: this sentence is more appropriate to the introduction or discussion.

Image 5 is a little confuse, in an attempt to put too much information in the same image,  some value are taken from all images.

Line 258; 260: The images resulted from the superimposed images of bright field and fluorescent are not sufficiently clear to draw conclusions about them.

Line 265: I am not able to see any punctual pattern of E- Cadherin immunostaining

Line 270:  In the presented image of TEX14 it is not possible to visualize the punctuated pattern.

Line 306: In my opinion, the explanation of what are cellular bridges should be clarified before in the manuscript.

Lines 322 and 326: Pan cytokeratins are markers of epithelial cells in general, many times used to confirm the epithelial origin of the tissue. Also needs a reference.

Lines 388 and 389: Please better explain in which way the intercellular communication influence the different cell phenotypes, please provide scientific support through literature references.

Line 420 and 421:  I am not quite sure that the techniques used by the authors is sufficient to affirm that there are different cell populations within the spheroid.

Figure 9: From my point of view it is very good to present a schematic view of the main findings in the spheroid formation, but this figure is not sufficiently self-explainable. The It seems to have a color code with a gradient of colors; however, this requires for sure a very detailed information to guide the readers.

-Future research directions may also be mentioned.

Figure S1: bad quality image, at least in the version provided for revisor´s download. However, from my experience with this cell line at 24 h cells start to compact to form the spheroid, then after forming the spheroid I admit that the diameter could increase after the formation, but during the formation the tendency id for compaction. Problem this could be related to the way the authors measured the diameters.

METHODS

Line 522: Please include the volume initially plated/well

Line 529: Why do the authors use the terminology aggregates and spheroids? Aggregates refer to small spheroids? Please clear this nomenclature.

Line 556: Please include references for the stem cell markers.

Line 560: Please include references for the TEX14 and the Pan-cytokeratin as markers for intercellular bridges.

Lines 568-570: Please provide more detailed information of the primary antibodies such as if they are monoclonal/ polyclonal, if monoclonal refer the clone. Also, it is extremely important refer the dilution of the primary and secondary antibodies. Maybe you could include all the information on a table, or accord to the authors’ preference, but the asked information must be clearly given.

- Clarify the number of hours that correspond to the overnight incubation

- What do the authors mean by extensive washes, this should be more descriptive, number of washed and time of washing.

579: Please include the dilution of the antibody

Lines 586-589: This information about what are TNTs is more adequate in the introduction section.

Line 592: Some author reported incompatibilities between RIPA buffer for sample collection and Bradford, because of the detergent of RIPA buffer, did you have this is consideration?

Line 599: Please include the dilutions of the primary antibodies.

Line 605: Please include the equipment where the chemiluminescence was read.

Line 612: Please report the dilution of the primary antibody.

Lines 620-624: In my opinion this information is more appropriate to the introduction section.

Line 627: Please include one reference.

Author Response

We thank the Reviewer for his/her work.

The manuscript has been revised taking into account the Reviewer’s suggestions. We have now rewritten both Introduction and Results by taking into account the Reviewer’s suggestions. We have also added requested details in Material and Methods.

(Results)

Line 75

WE have reworded the sentences.

Lines 84 and 85, 109

We have changed the text to better explain what happen in the RER cisternae and we have added references [numbers 28,29,36 in the “References”  section].

Moreover we have added detailed images.

Lines 98 and 99

Underlined format has been used

Lines 132 and 133

Done

Line 133

Information about the nucleus/cytoplasm ratio has been included in result an discussion sections.

Line 138

Done

Line 142

We have changed the photo that has been replaced with higher magnification one.

Line 143

We have changed the photo that has been replaced with higher magnification one. Moreover the description of large cytoplasmic stable bridges has been added.

Line 162-165

Done.

Line 165

According to several authors (Abounit and Zurzolo,2012; Rustom et al., 2014) TNTs display a lifetime ranging from minutres to several hours.

Reference has been inserted in the text.

Line 203-212

Done.

Line 213

The ultrastructural fibrillar aspect of amyloid material has been presented in a photo at higher magnification. Moreover we have added this information and the linked references in the text.

Line 230

Infomation about ROS and  a reference about the role of oxidative stress in cancer  are shifted in the introduction section [number 30 in the “References”  section].

Line 245

The sentence has been deleted

Line 247, 248

The sentence about ACTH/α-MSH has been shifted by taking into account the reviewer’s  suggestion.

Image 5

In the figure 5, the repertoire of the specific cellular events  is due to the necessity to show the complexity of the process.

Line 258, 260

The superimposed images (resulting from immulocalization and bright field), generally  required, are  proposed to better identify the involved area of spheroid.

Line 265

The punctate pattern of E-cadherin immunostaining is better visible in the photograph at higher magnification.

We are wondering if the Referees have seen the images in TIF or in another format (with loss of information) provided for the revision phase.

Line 270

The sentence has been deleted.

Line 306

The explanation of what are cellular bridges has been included in the text.

Line 322 and 326

We have added the reference  about Pancytokeratin expression as a differentiation state in tumours [number 41 in the “References”  ].

Line 388 and 389

Done

Line 420 and 421

We have reworded the sentence.

Figure 9

We have now added detailed information.

Conclusions have been added referring to future perspectives.

Figure S1

We have added several explanations

Methods

Line 522

Done

Line 529

We have explained in the text the difference between aggregate and small spheroid /spheroid

Line 556

We have added the references  about stemness markers [numbers 6, 37, 38 in the “References”].

Line 560

We have added the references  about TEX14 [numbers 52, 63, 67 in the “References” ], and about Pancytokeratin [number 41 in the “References”  section].

Line 560-70

-We have now added a table referring to used antibodies.

- we have clarify that usually, overnight incubation correspond to 14-16 hours.

-We added the number of washed and time of washing.

Line 579

Done

Line 586-589

The information about TNTs are shifted in the introduction section.

Line 592

We took in consideration that some authors reported incompatibilities between RIPA buffer for sample collection and Bradford, because of the detergent of RIPA buffer.The samples prepared with RIPA buffer are diluted in the cuvette for the measure of absorbance 1:500, so the final concentrations of SDS (0.0002%) and NP40 (0.002%) are absolutely compatible with the detergents tolerance (reported in the datasheet) of Bradford reagent (0.016% SDS; 0.5% NP40); moreover, the absorbance of our samples after correcting the blank absorbance with RIPA buffer has been measured.

Line 599

Done.

Line 605

Done

Line 612

The dilution of the primary antibody has not been reported because a plate with wells already coated (as part of the kit) has been utilized. The dilution is not reported in the datasheet.

Line 620-624

Information about ROS has been reported in introduction section.

Line 627

We have added the reference  about H2DCFDA [number 83 in the “References” ].

Reviewer 2 Report

MCF-7 is the subject of many investigations. It has been used extensively for 3D cultures. The authors  aim to close the gap in morpho-functional changes during aggregation and maturation. They aim to show evidence of tunneling nanotubes, amyloid fibril production and opening of larger cellular bridges that identify the sequential events leading to MCF7 spheroid formation.

The authors use many compound sentences that make it difficult to understand some of their sentences. The first sentence in the introduction and some subsequent sentences fit this description. They use multiple commas and are grammatically unclear. It is recommended that the authors keep their sentences clear and avoid unnecessary words. There are many places with unnecessary variations in font type and size.

Here are a few suggestions.

The understanding of the cell-cell interactions, as well as of the specific molecular events that participate in the dynamic way in breast cancer formation and growth, is of paramount importance.  Include the in the sentence.

To improve the faithfulness of solid tumour-based research and to decipher pathways, and morphofunctional changes that determine the histological complexity of this solid tumour, three-dimensional cell culture system are used. Replace faithfulness. Faithfulness is a weird word to use for tumors.

Line 47 MCF 7 spheroids can be generated (Include can)

Line 54 Here, (included comma after here)

Instead of - in the frame time of few hours,  - in a few hours

Line 79 -form aggregates different in size - that are different in size

Figures captions are overly wordy. Figure captions should be neutrally descriptive and not inferring what the reader should see.

Figure S1 is unclear 

Overall, analyzing MCF 7 spheroid formation at a deeper level provides some good insight into potential pathways to drug testing.

Author Response

We thank the Reviewer for his/her work.

The manuscript has been revised and Introduction and Results have been rewritten.

The figure legends have been modified

We have taken into account the suggested revisions.

Round 2

Reviewer 1 Report

The authors made an effort to incorporate most of the suggestions in the revised manuscript. The manuscript is now in conditions to be accepted.